# JASPAC: Japan Spastic Paraplegia Research Consortium

**DOI:** 10.3390/brainsci8080153

**Published:** 2018-08-13

**Authors:** Kishin Koh, Hiroyuki Ishiura, Shoji Tsuji, Yoshihisa Takiyama

**Affiliations:** 1Department of Neurology, Graduate School of Medical Sciences, University of Yamanashi, 409-3898 Yamanashi, Japan; jixin@yamanashi.ac.jp; 2Department of Neurology, Graduate School of Medicine, The University of Tokyo, 113-8655 Tokyo, Japan; hishiura@yahoo.co.jp (H.I.); tsuji@m.u-tokyo.ac.jp (S.T.)

**Keywords:** hereditary spastic paraplegias, JASPAC, SPG4, SPG3A, SPG31, SPG10

## Abstract

Hereditary spastic paraplegias (HSPs) are a group of neurodegenerative disorders characterized by weakness and spasticity of the lower extremities. HSPs are heterogeneous disorders that involve over 80 causative genes. The frequency of HSPs is estimated to be 10–100/1,000,000. With this background, the Japanese research group “Japan Spastic Paraplegia Research Consortium: JASPAC” was organized in 2006 to elucidate the molecular epidemiologies of HSPs in Japan and the molecular pathologies of HSPs. To date, the JASPAC has collected 714 HSP families and analyzed 488 index patients. We found 279 pathogenic variants or probable pathogenic variants of causative genes in the 488 HSP patients. According to our results, we found 178 families with autosomal dominant patients (65%), and 101 with autosomal recessive and sporadic patients (48%). We found 119 patients with SPG4, 17 with SPG3A, 15 with SPG31, 13 with SPG11, and 11 with SPG10. Other HSP genes were the cause in less than five patients. On the other hand, we could not find causative genes in 35% of the autosomal dominant patients, or 52% of the autosomal recessive and sporadic patients. We are now trying to find new causative genes and elucidate the molecular mechanisms underlying HSPs.

## 1. Introduction

Hereditary spastic paraplegias (HSPs) are a group of neurodegenerative disorders characterized by weakness and spasticity of the lower extremities. HSPs are classified clinically, or as to the inheritance pattern, or causative genes. According to clinical information, HSPs are classified into a pure form, which presents leg spasticity accompanied by bladder disturbance and disturbance of vibration sense, and a complicated form, which presents leg spasticity and a lot of complications such as ataxia, a thin corpus callosum, chorioretinal dystrophy, mental retardation, extrapyramidal signs, and peripheral neuropathy [1]. As to the mode of inheritance, HSPs are classified into autosomal dominant, autosomal recessive, X-linked, and mitochondrial inheritance. Complicated forms tend to exhibit autosomal recessive rather than autosomal dominant inheritance. To date, the causative genes or gene loci of HSPs, from SPG1 to SPG79, have been reported, and other hereditary diseases including Autosomal recessive spastic ataxia of Charlevoix-Saguenay (ARSACS), Chediak-Higashi syndrome, adrenoleukodystrophy, dystonia 4, and others, also exhibit leg spasticity as their one of the symptoms.

There are estimated to be 3–10 HSP patients per 100,000 individuals in Europe [2,3]. On the other hand, in Japan, there are estimated to be 0.2 cases per 100,000 individuals [4]. However, it has not been understood what subtypes of HSP exist in Japan. Furthermore, there is no causative treatment for HSPs so far. Therefore, the Ministry of Health, Labor and Welfare in Japan formed the Research Committee for Ataxic Disease to clarify the molecular epidemiology and pathogeneses of ataxic diseases including HSPs, and to identify targets of treatment. The Japan Spastic Paraplegia Research Consortium (JASPAC) is one of the groups of the Research Committee for Ataxic Disease. Here, we describe the JASPAC and its findings in detail.

## 2. Patients and Methods

The JASPAC was established in 2006. Its purpose was to collect genome resources for HSPs, to conduct comprehensive mutation analysis, to clarify the molecular epidemiologies of HSPs, and to clarify the molecular pathogeneses of HSPs. We collected HSP families using the family history or clinical information. Our HSP families include probands having affected parents or children, probands having an affected brother or sister, probands having parents of consanguineous marriage, and probands showing a complicated form with normal parents. According to our criteria, we have already recruited 714 patients. Written informed consent was obtained from all participants. We collected neurological examination results through our survey form.

Our previous flow chart for the comprehensive mutation analysis is presented in Figure 1. Patients determined to be autosomal dominant were subjected to direct sequencing of SPG4 and SPG31 at first. Patients with dementia or a thin corpus callosum and mental retardation were subjected to direct sequencing of SPG11 at first. Then, other patients and patients with no mutation of SPG4, SPG31, or SPG11 were subjected to rearrangement analysis by means of comparative genomic hybridization. The next analysis was resequencing microarray analysis. Finally, whole-exome analysis was performed. Exon capture was performed using a SureSelect All Exon V4+UTRs Kit and a V5+UTRs or V6+UTRs Kit followed by massively parallel sequencing using an Illumina Hiseq 2000 (100bp paired end). We aligned the exome data with a Burrows-Wheeler Aligner and extracted single nucleotide variations using Sequence Alignment/Map (SAM) tools. We checked for causative genes of neurodegenerative diseases. Furthermore, we compared the variants and their frequencies in an in-house control database consisting of 1261 subjects registered in Exome Aggregation Consortium (ExAC). We removed variants with a minor allele frequency of >0.2%, considering the disease frequency of HSP in Japan. Currently, we are performing whole-exome analysis before comparative genomic hybridization and resequencing microarray analysis. To date, we have completed the analysis in 383 patients.

## 3. Results

We found 224 patients (58%) with variants for 45 known HSP or neurodegenerative genes. These patients are listed in Table 1. The most frequent HSP was SPG4 (100 patients, 26%) among all HSP patients, followed by SPG31 (14 patients, 3.7%), SPG3A (13 patients, 3.4%), SPG11 (10 patients, 2.6%), and SPG10 (six patients, 1.6%). Other HSP genotypes, including SPG55, were found in five patients or less [5]. We found 25 causative genes in each one HSP family. Furthermore, we identified a lot of variants of other genes associated with different diseases, including *LYST* [6], *SACS*, *ABCD1*, *TUBB4A*, *ALS2*, *KCNA2*, *ABHD12*, *ANO10*, *CLCN1*, *CSF1R*, *GALC*, *GFAP*, *KCND3*, *MECP2*, *OPA1*, *PLA2G6*, *POLR3A*, *PSEN1*, *RNASEH1*, *SLC25A15*, *SYNE1*, *TYROBP*, *VPS13C*, and *WDR45.* These genes were reported as the cause of spasticity as one of the symptoms. Therefore, we classified these patients as complicated HSP patients accordingly.

We present clinical information on major pure form HSPs such as SPG4, SPG3A, SPG31, and SPG10 in Table 2. The average onset ages were 30.6, 14.3, 20.7 and 26.7 years old, respectively. Some of the patients exhibited ataxia, 9.0%, 7.7%, 7.0% and 16.7%, respectively. Neuropathy was also found in 31.0%, 0%, 29.0% and 33.3%, respectively.

## 4. Discussion

Our comprehensive mutation analysis revealed causative gene mutations in 224 HSP patients (58%). According to earlier reports [7,8], exome analysis revealed causative variants in 36% and 46% of their case series. Our HSP patients were collected carefully, which meant patients have to have a clear family history of autosomal dominant inheritance, severe phenotype or a family history that indicated autosomal recessive inheritance. This might have led to the high percentage of genetic diagnosis.

This study further expands the earlier molecular analyses of HSPs in Japan [7]. SPG4 is the most frequent cause of HSPs in Japan, as it is worldwide. It is the most frequent in 26% of the HSP families, which is similar to the earlier reported frequency [8]. Only three other HSPs, i.e., SPG31, SPG3A and SPG11, were found in over ten patients. HSPs with less than ten patients were caused by 44 genes. This indicates that HSPs are extremely heterogeneous diseases.

Furthermore, 24 genes, which comprise half the causative genes identified in this study, are not “SPG” genes, including the causative genes of Parkinson’s disease, amyotrophic lateral sclerosis, adrenoleukodystrophy, and others. These hereditary neurodegenerative disorders sometimes reveal upper motor neuron impairment. Furthermore, hereditary neurodegenerative diseases often involve several impaired systems, including the pyramidal tracts. Therefore, it is necessary to analyze genes comprehensively. JASPAC is succeeding in performing comprehensive gene analysis of HSP patients.

Our other purpose is to clarify the natural histories of HSPs and molecular pathogeneses of HSPs. To clarify the natural histories, we are tracking collected patients prospectively. Furthermore, we are now trying to find new causative genes and elucidate the molecular mechanisms underlying HSPs.

## Figures and Tables

**Figure 1 brainsci-08-00153-f001:**
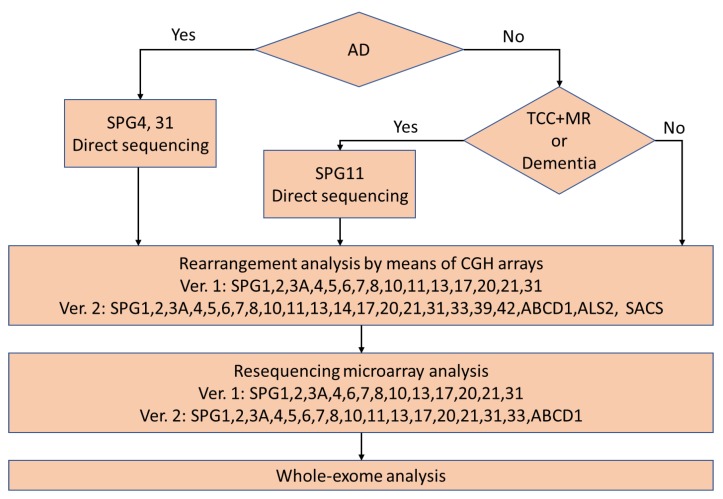
The earlier version of flow chart for our diagnostic procedure. In the current version, whole exome analysis is performed after excluding SPG4 (and SPG3A and SPG31 in patients with early-onset diseases) in familial patients with pure form. AD: autosomal dominant inheritance, TCC: thin corpus callosum, and MR: mental retardation.

**Table 1 brainsci-08-00153-t001:** Hereditary spastic paraplegia (HSP) genes found by the Japan Spastic Paraplegia Research Consortium (JASPAC).

SPG No. and/or Gene	Number	SPG No. and/or Gene	Number	SPG No. and/or Gene	Number
SPG4: *SPAST*	100	SPG12: *RTN2*	2	*CLCN1*	1
SPG31: *REEP1*	14	SPG28: *DDHD1*	2	*CSF1R*	1
SPG3A: *ATL1*	13	SPG35: *FA2H*	2	*GALC*	1
SPG11: *SPG11*	10	SPG48: *AP5Z1*	2	*GFAP*	1
SPG10: *KIF5A*	6	SPG78: *ATP13A2*	2	*KCND3*	1
SPG5: *CYP7B1*	5	*ALS2*	2	*MECP2*	1
*LYST*	5	*KCNA2*	2	*OPA1*	1
*SACS*	5	SPG6: *NIPA1*	1	*PLA2G6*	1
SPG9: *ALDH18A1*	4	*SPG7*	1	*POLR3A*	1
SPG15: *ZFYVE26*	4	SPG52: *AP4S1*	1	*PSEN1*	1
SPG46: *GBA2*	4	SPG54: *DDHD2*	1	*RNASEH1*	1
*ABCD1*	4	SPG55: *c12orf65*	1	*SLC25A15*	1
SPG2: *PLP1*	3	SPG56: *CYP2U1*	1	*SYNE1*	1
SPG30: *KIF1A*	3	SPG57: *TFG*	1	*TYROBP*	1
*TUBB4A*	3	*ABHD12*	1	*VPS13C*	1

**Table 2 brainsci-08-00153-t002:** Clinical information on common HSPs.

	SPG4	SPG3A	SPG31	SPG10
Number of patients	100	13	14	6
Average onset age (y. o.)	30.6	14.3	20.7	26.7
Ataxia (%)	9.0	7.7	7.0	16.7
Neuropathy (%)	31.0	0	29.0	33.3

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
