# Peer review of "JASPAC: Japan Spastic Paraplegia Research Consortium"

_brainsci, 2018, doi:10.3390/brainsci8080153_

Reviewer 1 Report

The authors provide an interesting and extremely concise overview about current knowledge on molecular bases of hereditary spastic paraplegia that alos allow to know the current frequency of each SPG sub-types in Japan.

My only criticism concerns the lack of precise molecular data : how the authors have classified the identified variants as benign, pathogenic or unknown significance. What is the rate of VUS with no clear conclusion.

The frequency of 58 % of causative genes identified through exome NGS is somewhat high compared to other studies in the field... please comment that point in the discussion.

Author Response

To the reviewer #1.

We have revised our manuscript (brainsci-322912) according to your useful comments and suggestions.

My only criticism concerns the lack of precise molecular data: how the authors have classified the identified variants as benign, pathogenic or unknown significance. What is the rate of VUS with no clear conclusion.

Reply.

We picked up all variants of the causative genes for HSPs. After that, we checked for the causative genes of other neurodegenerative diseases.

Furthermore, we compared the variants and their frequencies in an in-house control database consisting of 1,261 subjects and registered in ExAC. We removed variants with minor allele frequency >0.2% considering disease frequency of HSP in Japan. We have added the details under “Patients and Methods”.

The frequency of 58% of causative genes identified through exome NGS is somewhat high compared to other studies in the field…please comment that point in the discussion.

Reply.

We have added the following sentences: “According to earlier reports [7, 8], exome analysis revealed causative variants in 36% and 46% of their case series. Our HSP patients were collected carefully, which meant patients had to have a clear family history of autosomal dominant inheritance or severe phenotype or a family history that indicated autosomal recessive inheritance. This might have led to such a high percentage of genetic diagnosis.

Reviewer 2 Report

Kishin Koh et al. report the results of the genetic screening of 488 index patients of 714 HPS families collected by the Japan Spastic Paraplegia Research Cornsortium. They identified 279 causative alterations in different genes. Mutations in known HSP genes were found in 224 patient.

Although the authors spent great efforts to collect the samples and analyzed them, the manuscript does not further increase the knowledge on HPS to a larger extend at the current state.

Some points to address:

- All experimental steps including e.g. bioinformatics filter criteria in WES, sequencing technologies should be provided.

- Have segregation analyses been performed if a causative mutation has been identified?

- In the results section, line 74: “We found a lot of genes without SPG numbers…” phrasing is misleading and unclear what is meant. I think the authors mean, that they have identified mutations in other genes associated with different diseases. It would be good, if the authors would provide the detailed data on these genes and a list of the respective pathogenic mutations. If a mutation in a gene associated with another phenotype have been identified, did the authors reevaluated the clinical phenotype? Was there a HSP mimic or misdiagnosis?

Minor points:

- Phrasing is sometimes imprecise: a gene itself is never causative, it is a mutation within the gene.

- line 18: “…frequency of HSP is low…” this depends on the point of view

- line 34: “ hereditarily” is uncommon phrasing, maybe you replace it by “..by inheritance pattern”

- line 39 f.: “transmission” should be replaced by “inheritance”

- line 42: “…and other hereditary diseases also exhibit leg spasticity as one of ..…”  a more detail description of differential diagnoses would be good

- line 44 f: “population” should be replaced by “individuals”   

- line 46: “causative treatment”

- line 100: “… and so on.” A detailed description and discussion would be desirable

Author Response

To the reviewer #2.

We have revised our manuscript (brainsci-322912) according to your useful comments and suggestions.

Kishin Koh et al. report the results of the genetic screening of 488 index patients of 714 HPS families collected by the Japan Spastic Paraplegia Research Consortium. They identified 279 causative alterations in different genes. Mutations in known HSP genes were found in 224 patients.

Although the authors spent great efforts to collect the samples and analyzed them, the manuscript does not further increase the knowledge on HPS to a larger extend at the current state.

Some points to address:

1. All experimental steps including e.g. bioinformatics filter criteria in WES, sequencing technologies should be provided.

Reply

We have added bioinformatics information and tools under “Patients and Methods”.

2. Have segregation analyses been performed if a causative mutation has been identified?

Reply

We did not perform a segregation study. As you mentioned, the segregation study would be required in our future study.

3. In the results section, line 74: “We found a lot of genes without SPG numbers…” phrasing is misleading and unclear what is meant. I think the authors mean, that they have identified mutations in other genes associated with different diseases. It would be good, if the authors would provide the detailed data on these genes and a list of the respective pathogenic mutations. If a mutation in a gene associated with another phenotype have been identified, did the authors reevaluated the clinical phenotype? Was there a HSP mimic or misdiagnosis?

Reply

According to the reviewer’s comments, we have mentioned that we identified a lot of variants of other genes associated with different diseases, including LYST, SACS, ABCD1, and others. We confirmed that these genes caused spasticity. Therefore, we classified these patients as complicated HSP patients accordingly.

Minor points:

- Phrasing is sometimes imprecise: a gene itself is never causative, it is a mutation within the gene.

- line 18: “…frequency of HSP is low…” this depends on the point of view

- line 34: “ hereditarily” is uncommon phrasing, maybe you replace it by “..by inheritance pattern”

- line 39 f.: “transmission” should be replaced by “inheritance”

- line 42: “…and other hereditary diseases also exhibit leg spasticity as one of ..…”  a more detail description of differential diagnoses would be good

- line 44 f: “population” should be replaced by “individuals”  

- line 46: “causative treatment”

- line 100: “… and so on.” A detailed description and discussion would be desirable

Reply

According to your comments, we have rephrased or reworded minor points.

Reviewer 3 Report

The manuscript:"JAPSAC:Japan Spastic Paraplegia Research Consortium" raises a problem of epidemiology of HSP based on molecular studies in Japan. My major comment on it concerns the type of the article that looks as a short communication presenting the activity of a Consortium. If it is a research artlicle, some ammendments should be made:

Description of neurological inclusion criteria for patients.

Molecular analyses should be described in more details, for example: what is the whole-exome analysis? 

The authors could avoid the term: we found 279 causative genes, writing: we found 279 pathogenic vatiants (or mutations) or probably pathogenic variants in causative genes (abstract). Moreover, there is no information about mutation classification criteria and bioinformatic tools. 

It would be good to mention in the Results section that the analysis concerns only probands. In Table 1 45 genes are presented not 48.

Summarizing, the manuscript contains a lot of useful information about the clinical and molecular analyses of a largegroup of patients but needs a few corrections. 

Author Response

To the reviewer #3.

We have revised our manuscript (brainsci-322912) according to your useful comments and suggestions.

My major comment on it concerns the type of the article that looks as a short communication presenting the activity of a Consortium. If it is a research article, some ammendments should be made:

1. Description of neurological inclusion criteria for patients.

Reply.

Our neurological inclusion criteria for patients are given in the first paragraph of “Patients and Methods”, i.e., “Our HSP-----714 patients”.

2. Molecular analyses should be described in more details, for example: what is the whole-exome analysis?

Reply.

According to your comments, we have given details of the whole-exome analysis and criteria of extraction under “Patients and Methods”.

3. The authors could avoid the term: we found 279 causative genes, writing: we found 279 pathogenic variants (or mutations) or probably pathogenic variants in causative genes (abstract). Moreover, there is no information about mutation classification criteria and bioinformatics tools.

Reply.

We have rephrased as you mentioned, and have added information about criteria and bioinformatic tools.

4. It would be good to mention in the Results section that the analysis concerns only probands. In Table 1 45 genes are presented not 48.

Reply.

We apologize for the miscount.

Round  2

Reviewer 2 Report

Kishin Koh et al. addressed some of the points mentioned in my first comments. Unfortunately, the following one was not address in depth:

“- In the results section, line 74: “We found a lot of genes without SPG numbers…” phrasing is misleading and unclear what is meant. I think the authors mean, that they have identified mutations in other genes associated with different diseases. It would be good, if the authors would provide the detailed data on these genes and a list of the respective pathogenic mutations. If a mutation in a gene associated with another phenotype have been identified, did the authors reevaluated the clinical phenotype? Was there a HSP mimic or misdiagnosis?“

I still would like to see a list (table) of all neurodenegerative genes which were used to filter line 71-72 in the revised manuscript. Detailed molecular results (HGVS nomenclature, HGMD accession numbers, in silico predictions [if applicable] and MAF e.g. extracted from gnomAD browser) of all genes should be provided.

It is still not addressed if mutation carriers in neurodegenerative genes are HPS mimics or misdiagnoses. You make it to easy, to categorize all mutation carries as complicated HSP (line 87).

Author Response

Kishin Koh et al. addressed some of the points mentioned in my first comments. Unfortunately, the following one was not address in depth:

“- In the results section, line 74: “We found a lot of genes without SPG numbers…” phrasing is misleading and unclear what is meant. I think the authors mean, that they have identified mutations in other genes associated with different diseases. It would be good, if the authors would provide the detailed data on these genes and a list of the respective pathogenic mutations. If a mutation in a gene associated with another phenotype have been identified, did the authors reevaluated the clinical phenotype? Was there a HSP mimic or misdiagnosis?“

Response:

We wrote this paper as a review of JASPAC. As you know, we collected patients according to our criteria that patients having affected parents of consanguineous marriage, and probands showing a complicated form with normal parents. This decision was made by attending physician. Therefore, patients revealed spastic paraplegia with a lot of clinical manifestations. They looked like HSP, so they included mimic and misdiagnosis in them, we thought. However, according to our check, there were no genes associated with another phenotype. All diseases we found revealed spasticity, which indicated complicated HSP.

I still would like to see a list (table) of all neurodenegerative genes which were used to filter line 71-72 in the revised manuscript. Detailed molecular results (HGVS nomenclature, HGMD accession numbers, in silico predictions [if applicable] and MAF e.g. extracted from gnomAD browser) of all genes should be provided.

It is still not addressed if mutation carriers in neurodegenerative genes are HPS mimics or misdiagnoses. You make it to easy, to categorize all mutation carries as complicated HSP (line 87).

Response:

We screened over 150 genes which were known for neurodegenerative disorders. However, we wrote this paper as a review. Therefore, we would like to reveal only outline of our analysis.